# How multisite phosphorylation impacts the conformations of intrinsically disordered proteins

**Fan Jin**[1,2], **Frauke Gräter**[1,2]*

**1** Heidelberg Institute for Theoretical Studies, Heidelberg, Germany, **2** Interdisciplinary Center for Scientific Computing (IWR), Heidelberg University, Heidelberg, Germany

* frauke.graeter@h-its.org

**Data Availability Statement:** All relevant data are within the manuscript and its Supporting Information files.

**Funding:** F.G. acknowledges support from the German Research Foundation (DFG), project SPP

## Abstract

Phosphorylation of intrinsically disordered proteins (IDPs) can produce changes in structural and dynamical properties and thereby mediate critical biological functions. How phosphorylation effects intrinsically disordered proteins has been studied for an increasing number of IDPs, but a systematic understanding is still lacking. Here, we compare the collapse propensity of four disordered proteins, Ash1, the C-terminal domain of RNA polymerase (CTD2'), the cytosolic domain of E-Cadherin, and a fragment of the p130Cas, in unphosphorylated and phosphorylated forms using extensive all-atom molecular dynamics (MD) simulations. We find all proteins to show V-shape changes in their collapse propensity upon multi-site phosphorylation according to their initial net charge: phosphorylation expands neutral or overall negatively charged IDPs and shrinks positively charged IDPs. However, force fields including those tailored towards and commonly used for IDPs overestimate these changes. We find quantitative agreement of MD results with SAXS and NMR data for Ash1 and CTD2' only when attenuating protein electrostatic interactions by using a higher salt concentration (*e.g.* 350 mM), highlighting the overstabilization of salt bridges in current force fields. We show that phosphorylation of IDPs also has a strong impact on the solvation of the protein, a factor that in addition to the actual collapse or expansion of the IDP should be considered when analyzing SAXS data. Compared to the overall mild change in global IDP dimension, the exposure of active sites can change significantly upon phosphorylation, underlining the large susceptibility of IDP ensembles to regulation through post-translational modifications.

## Author summary

Intrinsically disordered proteins (IDPs) are a class of proteins that lack secondary and tertiary structures and instead explore a broad conformational ensemble. Their functions, from transcriptional regulation to signal transmission, are tightly regulated. IDPs are subject of extensive reversible post-translational modifications (PTMs), such as phosphorylation, methylation and glycosylation. Among these PTMs, phosphorylation is one of the most common and important PTMs. However, the mechanism of how phosphorylation affects the conformations and functions of IDPs remains unclear. To answer this question,

1782, and from the Klaus Tschira Foundation. F.J. is grateful to funds from BIOMS program of Heidelberg University. The authors acknowledge support by the state of Baden-Württemberg through bwHPC and the German Research Foundation (DFG) through grant INST 35/1134-1 FUGG, and by the Gauss Centre for Supercomputing e.V. (www.gauss-centre.eu) for funding this project by providing computing time through John von Neumann Institute for Computing (NIC) on the GCS Supercomputer JUWELS at Jülich Supercomputing Centre (JSC). The funders had no role in study design, data collection and analysis, decision to publish, or preparation of the manuscript.

**Competing interests:** The authors have declared that no competing interests exist.

we have performed extensive all-atom molecular dynamics simulations for four representative IDPs: Ash1, E-Cadherin, CTD2' and p130Cas in their unphosphorylated and phosphorylated forms. Our results showed that all IDPs undergo a mild change upon multisite phosphorylation, which is V-shaped: phosphorylation moderately expands neutral or overall negatively charged IDPs and shrinks positively charged IDPs. More importantly, in two of these IDPs, only two biologically relevant phosphorylation sites suffice to render the adjacent negatively charged active site significantly more exposed to the environment, which implies a higher probability to interact with other binding partners.

## Introduction

Intrinsically disordered proteins (IDPs) lack stable secondary and three-dimensional structures, and instead feature a broad ensemble of rapidly interconverting conformations. Their conformational plasticity is critical for their biological functions [1,2]. Post-translational modifications alter the conformational ensemble and thereby the function of IDPs [3,4]. Phosphorylation is a modification which introduces a bulky moiety carrying negative charges to otherwise neutral residues (commonly Thr/Ser/Tyr), and thus has the potential to strongly alter the electrostatic interactions within and between IDPs. With IDPs sampling a largely flat free energy landscape, such a perturbation of the electrostatic interactions has the potential to greatly shift an IDP's conformational ensemble with regard to both local and global structural propensities. Furthermore, IDPs are heavily involved in formation of various kinds of higher-order assemblies, and it is crucial to uncover major driving forces of phosphorylation-based regulation of biomolecular condensates [5].

Under the simple hypothesis that electrostatics dominate, phosphorylation of overall negatively charged IDPs is supposed to result in more extended conformations on average while phosphorylation of positively charged IDPs is supposed to result in more compacted conformations. In this view, the dominant driving forces causing a shift of the IDP conformational ensemble across its characteristic flat energy landscape are repulsive/attractive electrostatic forces between the negatively/positively charged residues and the phosphorylated residues (Fig 1). Recently, a number of studies allowed first very valuable insights into the effect of phosphorylation on IDP structure [6–9]. Small Angle X-ray Scattering (SAXS), nuclear magnetic resonance (NMR), and fluorescence resonance energy transfer (FRET) are among the most commonly used experimental methods for IDP structural and dynamics studies [10], also when it comes to studying phosphorylation [11–15]. SAXS delivers a radius of gyration, $R_G$, as a measure of the global IDP collapse propensity. Recently, an extended Guinier analyses was developed to increase the q×Rg range and thus enhance the reliability of the Guinier analyses of SAXS data [16]. Progress has also been made in directly comparing SAXS profiles obtained from Molecular Dynamics (MD) simulations to experimental SAXS data to address the problem of solvation inherent to SAXS data [17,18]. Recent force field developments for IDPs include the Tip4p-D water model for increased dispersion with water [19], the application of the Kirkwood-Buff forcefield for IDPs [20,21], and modifications of previous protein force fields, namely Charmm36m [22] and a99SB-disp [23], to accurately treat both IDPs and folded proteins. However, charge-charge interactions in current force fields have far less been in the focus for the case of IDPs. On one hand, it was realized that standard forcefields are prone to over-stabilize salt bridges [24]. Consequently, an Amber99SB* ILDN-DERK forcefield based on Amber99SB* ILDN [25] with optimized charges on Asp, Glu, Arg and Lys has been suggested [26]. On the other hand, both the overall Net Charge per Residue (NCPR) as well as the

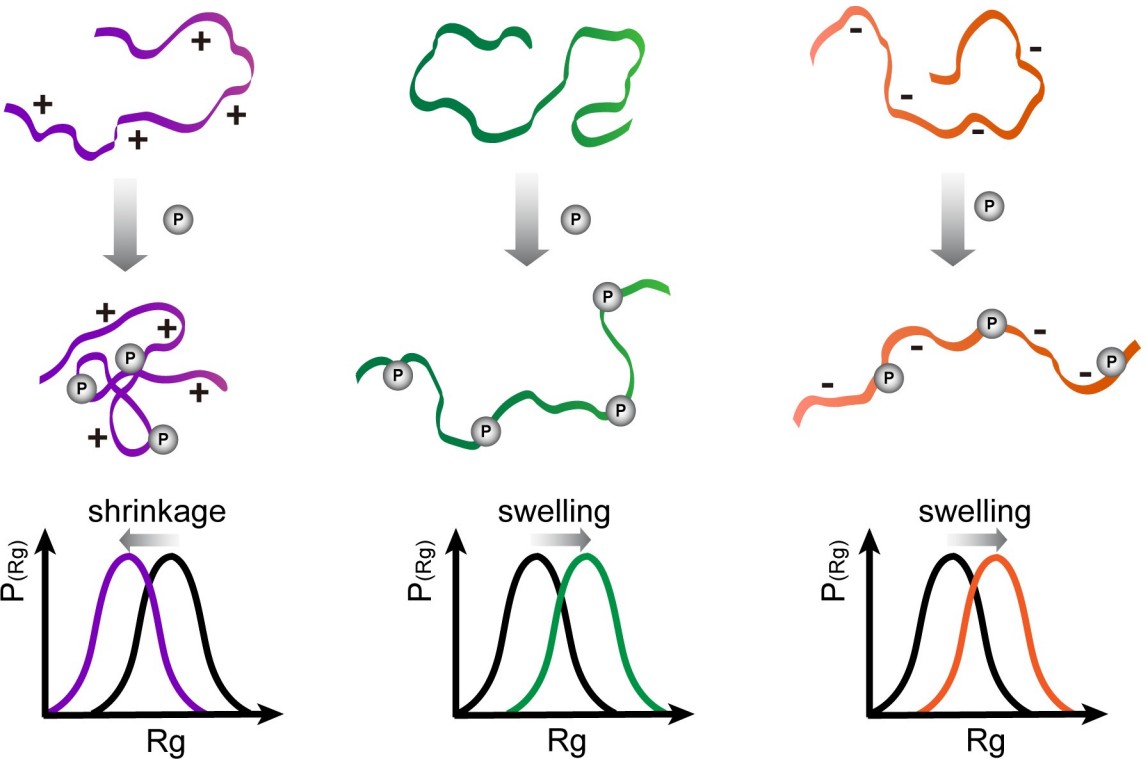

**Fig 1. Scheme of the potential effect of phosphorylation on the global conformation of IDPs.** Assuming electrostatics to be the major determinant, an overall neutral or negatively charged IDP swells upon phosphorylation, while a positively changed IDP shrinks accordingly.

distribution of opposite charges along the linear sequence can strongly affect the global dimensions of IDPs, as shown by all-atom simulations with implicit solvent [10,27,28]. More recently, the conformational changes upon phosphorylation of a short IDP fragment of statherin were investigated by comparing the MD simulations using AMBER ff99SB-ILDN and CHARMM36m force fields with SAXS and circular dichroism experiments [29]. The two force fields performed less well for the phosphorylated than for the unphosphorylated states of statherin and also showed differences between each other with regard to secondary structure propensities.

Here, to answer the question how phosphorylation affects the conformational ensemble of IDPs and illuminate a relationship between the governing rules of conformational change and biological function, we conducted extensive all-atom explicit solvent molecular dynamics simulations on four representative IDPs known to undergo multi-site phosphorylation. Namely, we chose Ash1, the carboxy-terminal domain (CTD) of RNA polymerase II (CTD2'), the cytosolic domain of E-cadherin and p130Cas, and from each of these IDPs a fragment of similar sequence length comprising multiple phosphorylation sites (Table 1). Ash1 is a transcription factor that regulates Saccharomyces cerevisiae's mating type switching. Previous work reported the 10 phosphorylation sites Ash1$^{420-500}$ to undergo a very small conformational change (change in $R_G$ of 1 Å) upon phosphorylation [6]. CTD2' is a large disordered subunit, the phosphorylation cycle of which correlates with the transcription cycle and regulation of mRNA [7]. Both phosphorylation of Ash1 and CTD2' are supposed to be affected by multiple prolines in the sequence due to the higher rigidity of the backbone [6,7]. E-cadherin is a cell-cell receptor protein in adherens junctions. Phosphorylation of its cytosolic disordered domain

**Table 1. Primary sequences of IDPs investigated in MD simulations: Ash1, CTD2' of RNA polymerase II, the cytosolic domain of E-Cadherin, and p130Cas with UniProt IDs shown below the name of the IDPs.** The positively and negatively charged residues are colored in cyan and red, respectively. The phosphorylation sites are underlined in purple. For E-Cadherin and p130Cas, the most biologically relevant two phosphorylation sites are shown in larger font size. The net charges of fully unphosphorylated and phosphorylated forms of these IDPs are shown in the right panel.

| Protein | Sequence | net charge unphosphorylated | net charge phosphorylated |
|---|---|---|---|
| **Ash1** P34233 | $^{420}$SASSSPSPSTPTKSGKMRSRSSSPVRPKAYTPSPRSPNYHRFALDSPPQSPRRSSNSSITKKGSRRSSGSSPTRHTTRVCV$^{500}$ | +15 | -5 |
| **CTD2'** AAA28868.1 | $^{1659}$FAGSGSNIYSPGNAYSPSSSNYSPNSPSYSPTSPSYSPSSPSYSPTSPCYSPTSPSYSPTSPNYTPVTPSYSPTSPNYSASPQ$^{1741}$ | 0 | -20 |
| **E-Cadherin** P12830 | $^{735}$AVVKEPLLPPEDDTRDNVYYYDEEGGGEEDQDFDLSQLHRGLDARPEVTRNDVAPTLMSVPRYLPR$^{800}$ | -9 | -25 |
| **p130Cas** P56945 | $^{631}$SIQSRRLPSPPKFTSQDSPDGQYENSEGGWMEDYDYVHLQGKEEFEKTQKELLEKGSITRQGKSQL$^{696}$ | -4 | -16 |

promotes its cell surface stability and adhesion by regulating the binding of adaptor proteins [30]. p130Cas, or Breast cancer anti-estrogen resistance protein 1, is an adaptor protein of Src family proteins. Its interactions with multiple binding partners are regulated by multisite phosphorylation [31], which in turn is enhanced by force-induced stretching of the IDP [32].

Through MD simulations, we here observed that phosphorylation of IDPs leads to a shrinkage of IDPs in case their total net charge is positive, and to an expansion of those IDPs with a negative net charge, directly following what simple electrostatics would predict. We found, however, that IDP-tailored forcefields overestimate this effect. An increased NaCl concentration of 350 mM that screens electrostatics in the simulations was able to best mimic the experientially observed ensembles. Furthermore, we found that experimental SAXS data can hide a phosphorylation-induced change in collapse propensity as changes in SAXS-derived $R_G$ also include solvation. Based on our findings, we propose a "protect-attack" mechanism as a general function of phosphorylation/dephosphorylation of IDPs.

## Results and discussion

### MD simulations suggest shrinkage of Ash1 and expansion of CTD2' upon phosphorylation

We started out with MD simulations, using AMBER99-sb*-ILDN forcefield with TIP4P-D water [19,25], of Ash1 and CTD2' of RNA polymerase II in multi-sites phosphorylated and unphosphorylated states, as the conformational ensembles of these two IDPs have been also experimentally characterized previously [6,7]. We observed that phosphorylation reduced the mass-weighed radius of gyration, $R_G$, of Ash1/pAsh1 by 10 ± 4 Å and increased that for CTD2'/pCTD2' by 14 ± 6 Å at a NaCl concentration of 100 mM (Figs 2, S1 and S2). Both changes can be straightforwardly explained by the IDPs' net charges: phosphorylation reduces the overall electrostatic repulsion along Ash1 with net charge +15, leading to global contraction. Instead, it further adds negative charges to CTD2' with zero net charge, leading to electrostatic repulsion and swelling. In these simulations, phosphorylated Serine and Threonine were fully deprotonated and carried a net charge of -2, which is their prevalent protonation state at pH 7.5 (physiological condition, as used in Ref [6] and [7]), according to the typical pKa of phosphoesters of 5.5 to 6.0 [7,33]. For singly protonated phosphates as prevalent at for 2.2<pH<6.0, we also observed a shrinkage of Ash1 and CTD2', albeit with a change in $R_G$ of ~5 Å less pronounced (S1 and S2 and S4 Figs). A phosphomimetic Ash1 with phosphorylation sites mutated to Glutamate, as used previously in implicit solvent simulations [6], closely followed the trend of singly protonated phosphorylated Ash1 but not of the deprotonated (phosphates with -2 charge) form (S1 Fig). This underlines that IDPs are very sensitive towards

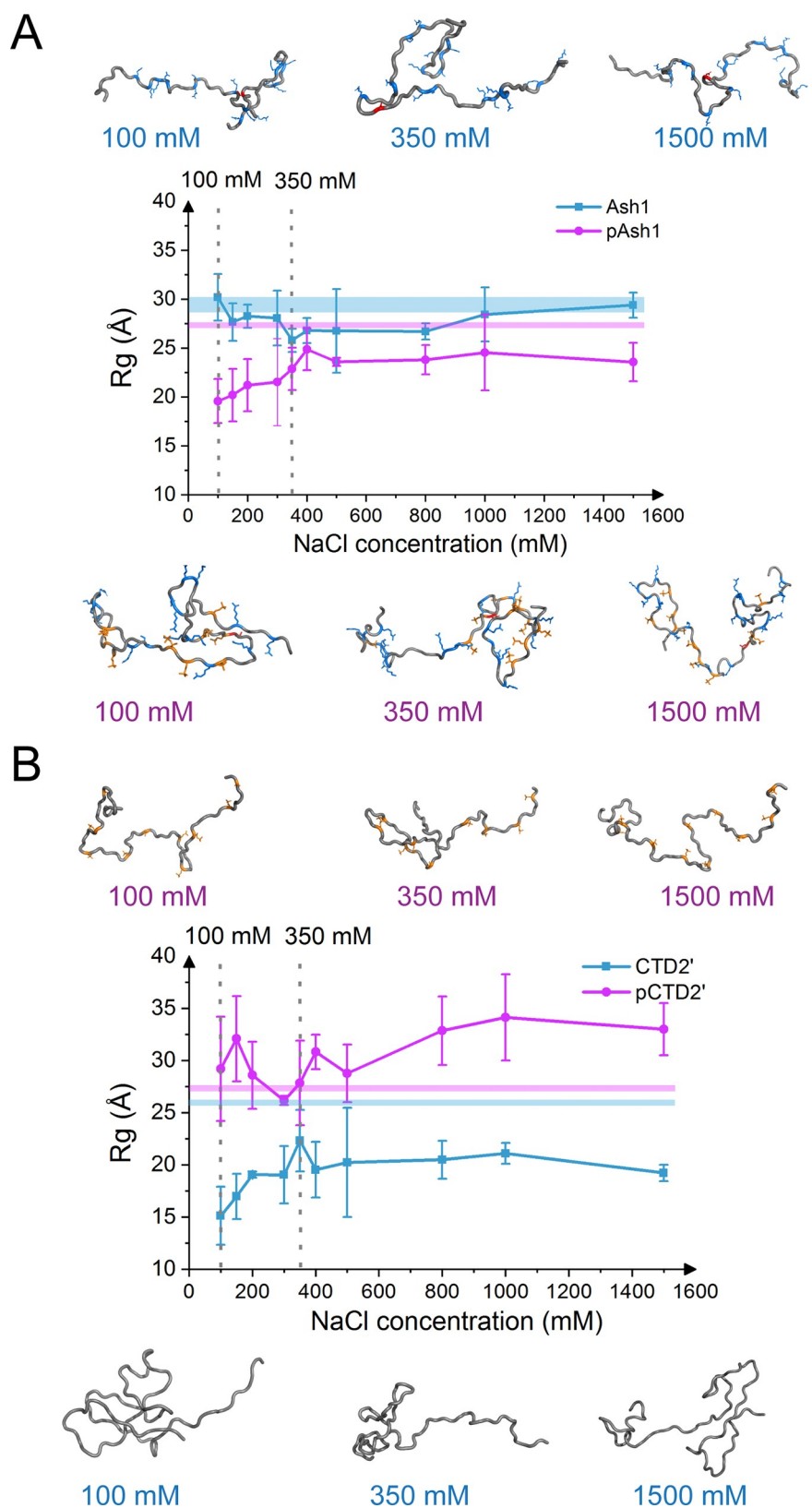

**Fig 2. Higher ion strength attenuates phosphorylation-induced shrinkage of Ash1 and expansion of CTD2'.**
Global conformation changes, measured by Rg, the mass-weighted radius of gyration, of Ash1 (top) and CTD2'

(bottom) for unphosphorylated (cyan) and multi-sites phosphorylated (purple) forms under different salt concentrations. Experimental values with standard deviations are shown by horizontal lines. Results for 100 mM and 350 mM are emphasized by dashed vertical lines. Representative conformations of MD simulations under 100, 350 and 1500 mM are shown as cartoon. The phosphorylated residues are colored in orange, while the positively/negatively charged residues are colored in blue/red, respectively.

changes in charge states of amino acids e.g. by pH changes, and that care must be taken when using phosphomimetic glutamate to study phosphorylation effect for IDPs in experiments or simulations.

## IDP-adequate force fields overestimate the charge effect

In contrast to the swelling of CTD2' and the collapse of Ash1 upon phosphorylation, SAXS results showed that global conformational changes of Ash1 and CTD2' upon phosphorylation are largely absent or at most minor (change in $R_G$ of 1–2 Å), according to the SAXS profiles and Guinier analyses [16]. Given that the current protein force fields have been shown to overstabilize electrostatic interactions [24], we hypothesized that the MD simulations overestimate the effect of electrostatic interactions. To test this, we performed additional simulations of the two IDPs and their phosphorylated forms at higher salt concentrations up to 1.5 M which screen electrostatic interactions. As expected, NaCl reduced the change in mass-weighed radius of gyration for both Ash1/pAsh1 and CTD2'/pCTD2', i.e. weakened the phosphorylation effect. This suggests that the simulations at 100mM indeed feature an overstabilization of intramolecular salt bridges, in agreement with a previous MD study on salt bridges in folded proteins [24]. Our finding is also in line with the observation of an over-compacted phosphorylated state of statherin stabilized by a salt bridge [29]. In the case of IDPs considered here, the overstabilization leads to an overestimation of the effect of phosphorylation on the highly flexible and broad conformational ensembles of Ash1 and CTD2'. From the comparisons, we found MD simulations under 350 mM NaCl to yield the most similar collapse propensities of the phosphorylated and unphosphorylated states, namely 2–3 Å collapse of Ash1 and 4–5 Å expansion of CTD2' upon phosphorylation. We consider this salt concentration (more generally concentrations in the 300-400mM range) to best mimic the experimentally observed behaviors of Ash1/pAsh1 and CTD2'/pCTD2' (Fig 2). Indeed, we observe fewer salt bridges within pAsh1 for the range of 300-400mM NaCl compared to lower salt concentrations, suggesting that higher salt indeed reduces the overstabilization of salt bridges inherent to the force field (S2 Fig).

   To more quantitatively validate the MD simulations, we back-calculated chemical shifts for Ash1/pAsh1 and observed a good agreement with experimental values (Figs 3 and S3 and S1 Table, chemical shift RMSD of 0.4–0.9 ppm for carbons and 1.6–2.3 ppm for nitrogen). The overall RMSDs from experimental chemical shift values are largely robust with regard to salt concentrations, with a slight decrease in RMSD for hydrogen chemical shifts in the range of 300-400mM compared to lower salt, corroborating that higher salt improves the overall quality of the predicted Ash1/pAsh1 ensembles (S1 Table and S3 Fig). In addition, MD-derived SAXS curves of Ash1/pAsh1 and CTD2'/pCTD2', calculated taking the solvation explicitly into account, are in line with experimental curves (Figs 4 and S5). We conclude that using 350mM NaCl yields conformational ensembles of the two IDPs and their phosphorylated forms which agree with SAXS and NMR data and thus resemble the experimental ensemble. Our MD data also suggest that IDPs, sampling conformations on a very flat free energy landscape, can be very sensitive towards ionic strength if they feature a high abundance of charged residues or phosphorylated residues. This, however, becomes only evident at low ionic strength conditions. Lowest ionic strengths used in experiments for Ash1/pAsh1 and CTD2'/pCTD2'

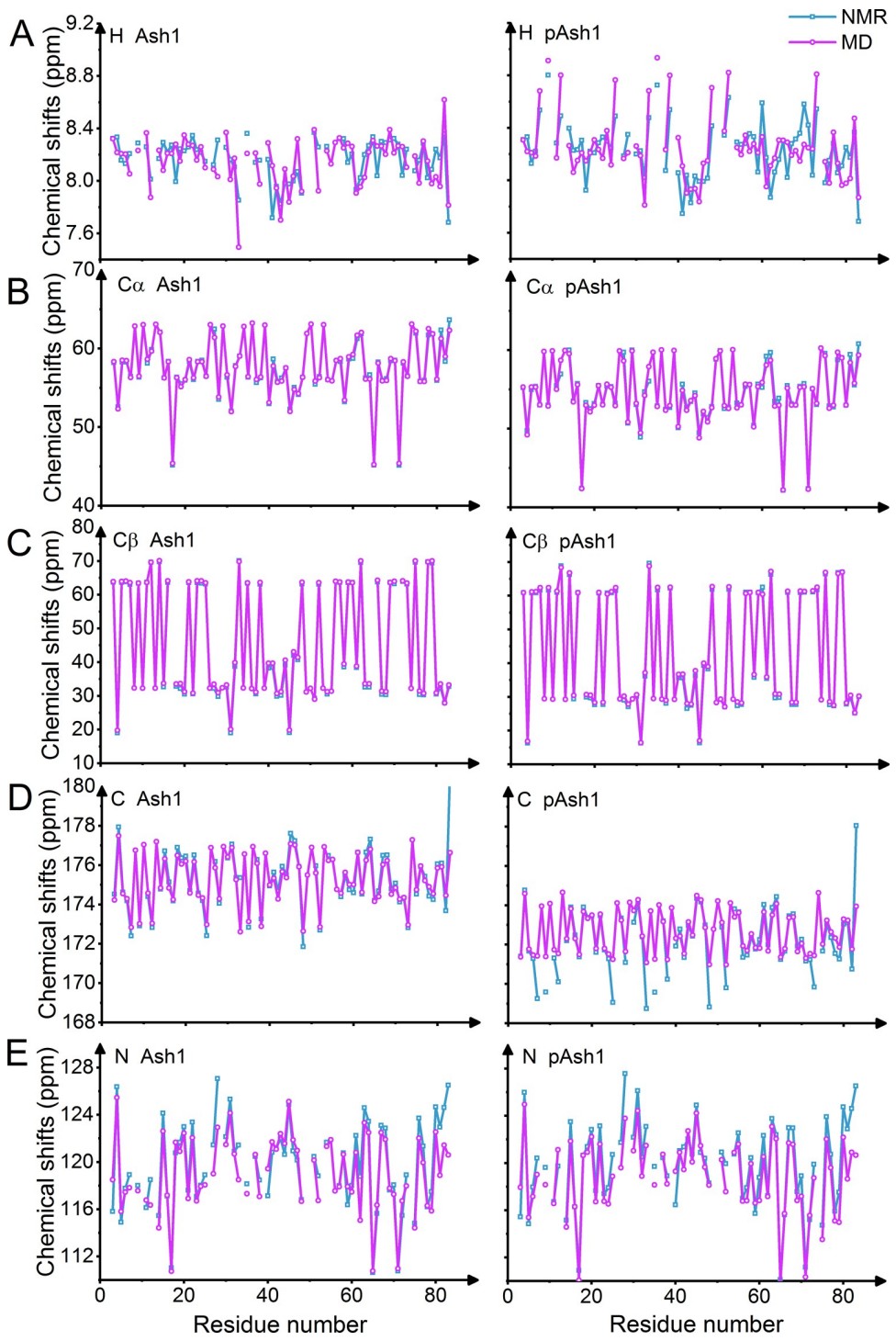

**Fig 3. Validation of MD simulations by comparisons to NMR data.** Comparisons of simulation-derived (purple, 350 mM NaCl) and experimental chemical shifts (cyan, 150 mM NaCl) for unphosphorylated (left panel) and phosphorylated (right panel) forms of Ash1 show very good agreement. Note that the experimental values for some residues were not available. Chemical shifts are for the atoms: A H, B Cα, C Cβ, D C, E N. Experimental data from REF [6].

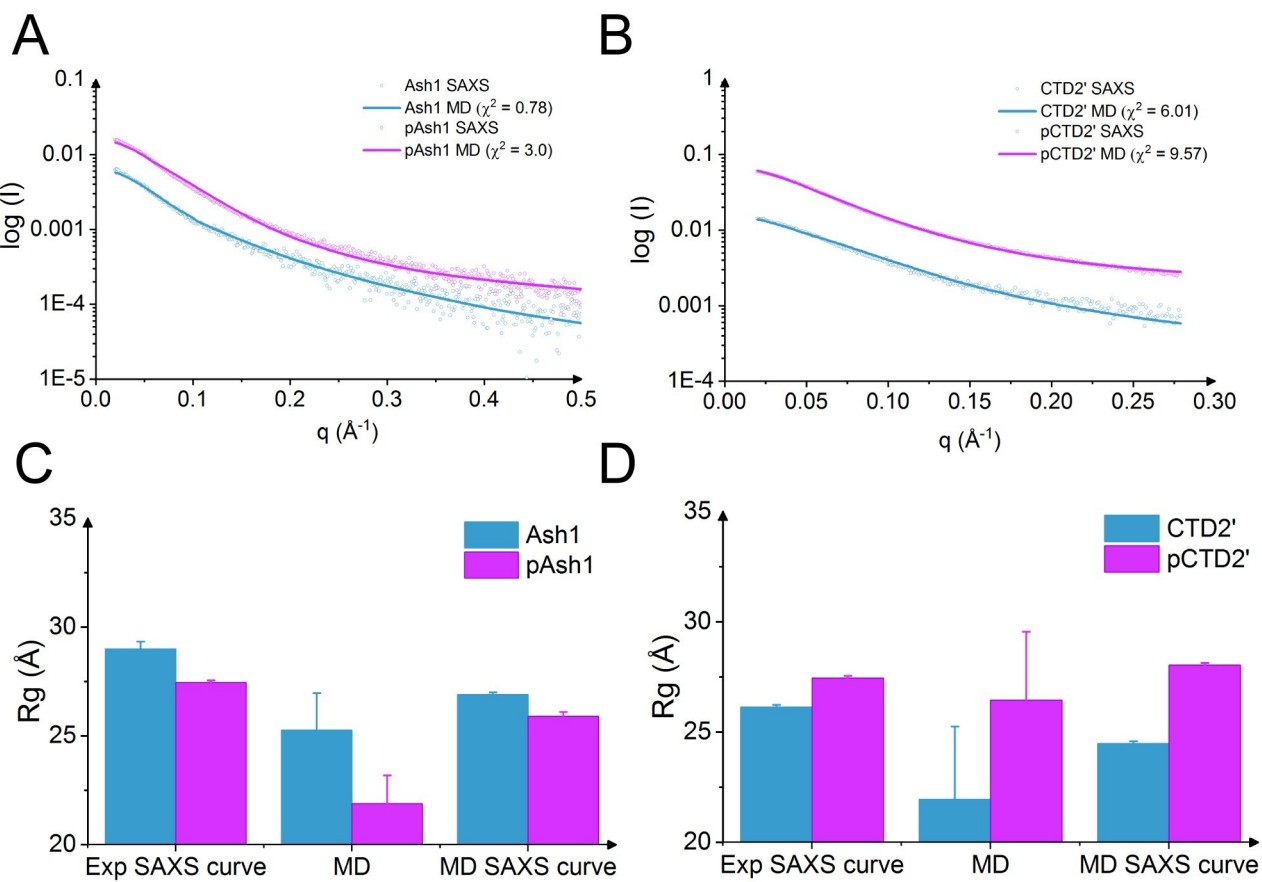

**Fig 4. Validation of MD simulations by comparison to SAXS data.** Reversely calculated SAXS curves for Ash1/pAsh1 (A) and CTD2'/pCTD2' (B) obtained at 350 mM NaCl were compared with experimental curves (150 mM NaCl). The radius of gyration from SAXS experiments using extended Guinier analysis ('Experimental SAXS curve'), directly calculated from MD simulations ('MD'), and calculated from MD-based SAXS data using Guinier analysis ('MD SAXS curve') were compared for Ash1/pAsh1 (C) and CTD2'/pCTD2' (D).

(including buffer) have so far been 125 mM or 130 mM, respectively [6,7], where electrostatic screening already is at play. We predict the change in collapse propensity upon phosphorylation to be more evident at even lower ionic strength, e.g. when only using buffer (50-75mM in previous experiments) and no additional salt. Such experiments could give new insight into the role of salt bridges in shaping the conformational ensemble of charged and/or phosphorylated IDPs.

Next, we asked whether other protein force fields also overestimate the charge effect. To this end, we performed additional MD simulations for Ash1/pAsh1 and CTD2'/pCTD2' with the Charmm36m [22], the a99SB-disp [23], and the Amber99SB* ILDN-DERK forcefields [26,25]. In the case of Charmm36m, irrespective of the IDP and the phosphorylation, we observe a strong overcollapse of the IDP which is not in line with the SAXS data (S6 Fig). For the two Amber force field variants, we recovered the same trend as for AMBER99-sb*-ILDN with TIP4P-D with highly similar changes in $R_G$ (S7 and S8 Figs), suggesting that the overestimation of the change in global collapse propensity upon phosphorylation is not unique to one specific IDP force field. In fact, a major emphasis in force field development for IDPs has been on dispersion and solvation [18,20,21] and local dihedral propensities [22] while the protein charge-charge interactions have remained untouched. We thus expect the too strong sensitivity of the

global dimensions of IDPs towards phosphorylation to be a general problem. Recently, Lindorff-Larsen and co-workers modified the charges of Glu, Asp, Arg, and Lys in Amber99SB* ILDN to overcome the overstabilization of salt bridges, resulting in the Amber99SB* ILDN--DERK forcefield [25,26]. However, these parameters also did not overcome the limitation, and again, only 350mM of salt resulted in conformational ensembles of the phosphorylated and unphosphorylated Ash1 protein in line with experiments (S7 and S8 Figs).

## Phosphorylation effects hydration of Ash1 and CTD'

According to Guinier analyses of the experimental SAXS data, there is only a small change (~2 Å) upon phosphorylation for Ash1 and CTD2' along the direction expected from the initial net charge (Fig 4C and 4D) [6,7]. We recover the same trend in MD simulations: shrinkage of Ash1 and swelling of CTD2' upon phosphorylation. However, the change in experiments is smaller than the change observed in MD simulations according to the directly calculated $R_G$ (~4–5 Å), even at 350mM. We asked if the solvation explains the remaining discrepancy. We again conducted Guinier analyses on the simulation-derived SAXS curve, and interestingly, the $R_G$ from the simulation-derived SAXS curve showed a trend very similar to the $R_G$ from Guinier analysis of experimental SAXS data, namely a ~2 Å shrinkage for Ash1 and expansion of CTD2'. The simulation-derived SAXS data takes the solvation shell into account, thus measuring not only the protein by itself but the radius of gyration with contributions from ions and solvent, in direct analogy to the experiments [18,34]. Thus, the $R_G$s of only the protein ensemble as observed in MD ('MD') are by definition smaller. More importantly, the hydrodynamic radius ('MD SAXS curve' in Fig 4C and 4D) changes much less upon phosphorylation than the actual protein $R_G$ ('MD'). This implies that the SAXS-derived RG includes a larger solvation shell for the phosphorylated than for the unphosphorylated ensembles. We note that the standard errors of the MD $R_G$s are large due to the high flexibility and long equilibration time scale of the IDPs. Yet, both IDPs show similar trends in this comparison, corroborating this finding. We conclude that phosphorylated and unphosphorylated proteins can feature different solvation shells, which is not surprising, given the highly charged phosphate moieties. This effect in the case of Ash1 and CTD2' hides global conformational changes of the IDP: while SAXS Guinier analyses show only minor changes (~2A) the actual changes of the protein ensemble, disregarding solvation, underlying this SAXS data are larger (~4-5A). It is likely that such scenario is also at play for other IDPs or other changes in charge state, such as acetylation or protonation state changes.

## Net charge explains changes in collapse propensity upon phosphorylation

We next asked if changes in IDP dimensions upon phosphorylation can be predicted merely from sequence. To this end, we extended the set of simulated IDPs to other IDPs known to undergo multi-site phosphorylation in vivo, namely p130Cas and E-Cadherin. We calculated the $R_G$s of unphosphorylated states, partially phosphorylated states (two most biologically relevant P-sites, only for p130Cas and E-Cadherin, Table 1), and fully phosphorylated states in equilibrium MD simulations, again using the AMBER99-sb*-ILDN force field [25] with TIP4PD water. We used 350mM NaCl, as this salt concentration yielded conformational ensembles in close agreement to experiments. We obtained a very good correlation of the normalized $R_G$ [35] with net charge per residue, NCPR (Fig 5). The positively charged IDP Ash1 shrinks towards a net charge of zero upon phosphorylation, while uncharged or negatively charged IDPs show an expansion upon phosphorylation, resulting in a V-shape of the collapse propensity along NCPR. We also compared these results to MD simulations of 100 mM NaCl, which effectively yield an overestimation of the charge effect (dashed lines in Fig 5), and

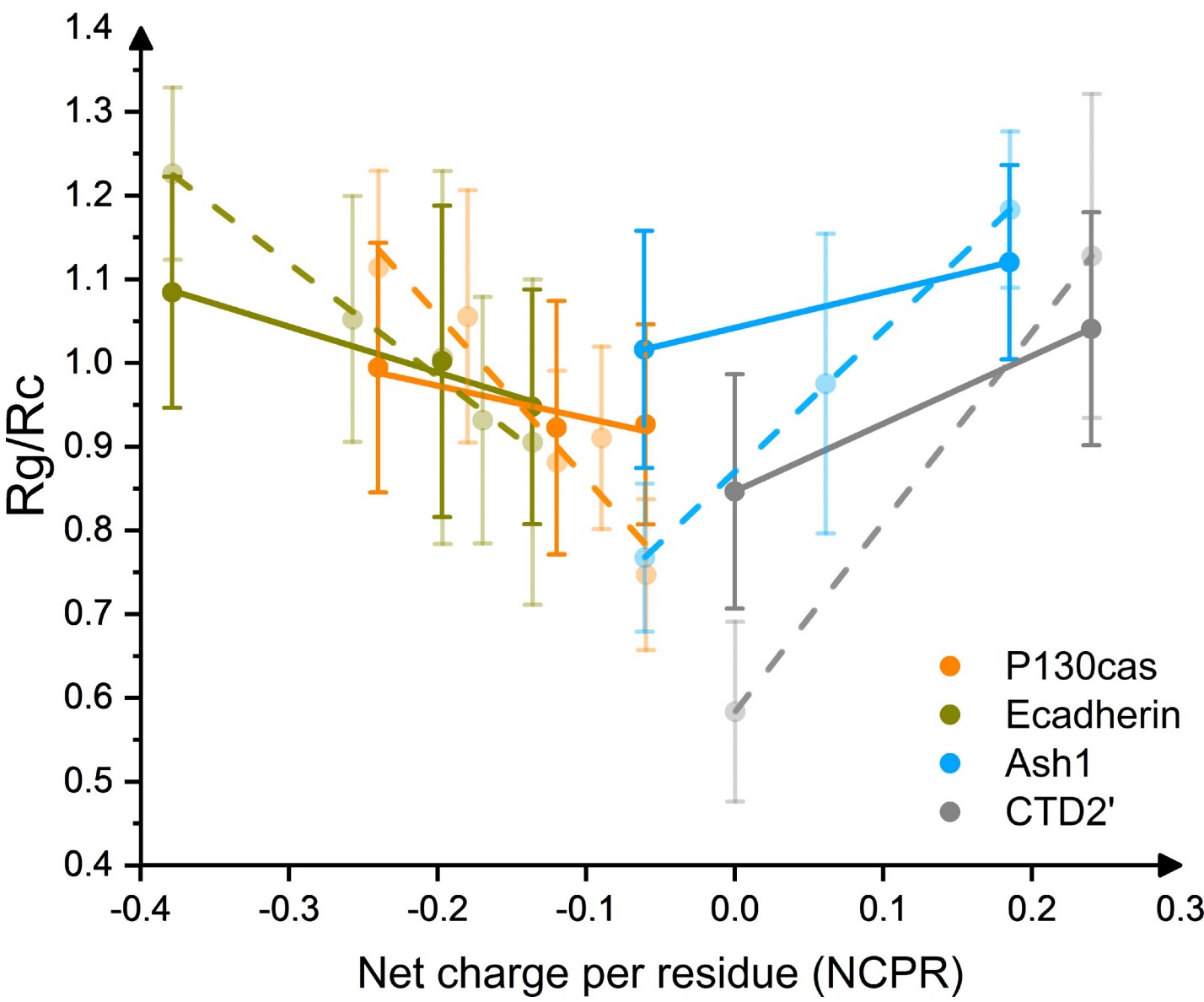

**Fig 5. Correlation of Net charge per residue with radius of gyration, normalized to random coil dimension ($R_c = R_0 \times N^{0.588}$ [35], where $R_0$ is a constant and N is number of residues of IDP) for Ash1/pAsh1 (light blue), CTD2'/pCTD2' (gray), E-Cadherin/pE-Cadherin (dark yellow) and p130Cas/pp130Cas (orange), respectively, under 350 mM (solid) and 100 mM (transparent) NaCl.** Linear fittings are shown in solid (350 mM NaCl) and dashed (100 mM NaCl) line. The CTD2'/pCTD2' states are shown in mirror for clarity.

observed a steeper V-shape, as expected. We predict such stronger effects to become evident in conditions of less screening, i.e. at very low or zero ionic strength. The correlation of the normalized $R_G$ with the absolute value of NCPR is significant, with a slope from linear fits significantly different from zero (p-values <0.05, S9 Fig). We also correlated the changes in $R_G$ with other parameters from sequence, such as 'Coulomb force and potential' (S10 Fig). Interestingly, these more involved parameters yield a correlation which is less strong than the one with NCPR. This implies that the IDPs considered here feature patterns of charged residues and phosphorylation sites along sequence which resemble each other, such that a single parameter, as simple as their net charge, can capture their distribution.

We found NCPR to be a very good indicator of the change in global IDP dimension upon phosphorylation, which points to an important role of salt bridges in shaping the conformational ensemble. We analyzed the salt bridge propensties and found salt bridges involving the phosphate groups, but not the other salt bridges, to strongly correlate with the overall shrinking and swelling behavior described above (Fig 6). For example, the positively charged IDP Ash1 shrinks upon phosphorylation, involving—and likely driven by—a significant increase of salt bridges (Fig 6A). Instead, salt bridges of negatively charged E-Cadherin only increase slightly upon phosphorylation (Fig 6B), as a strong repulsive force from negatively charged residue oppose their formation in pE-Cadherin. We observed an only limited influence of IDP phosphorylation on the number of D/E–R/K salt bridges for Ash1 (Fig 6C), suggesting direct phosphate-D/E/R/K interactions to drive the observed changes in IDP dimensions in the case of positively charged IDPs (Fig 6E). When it comes to E-Cadherin, D/E-R/K interactions were strongly reduced upon phosphorylation because of the expansion of the conformations (Fig 6D), and only partly compensated by newly established salt bridges involving phosphates (Fig 6F). Salt bridges thus play very important roles in shaping the conformations of IDPs (Fig 6G).

## Unshielding of protein interaction sites by phosphorylation

Phosphorylation of IDPs mediates critical biological functions. As shown previously and also in this study, in experiments and simulations, IDPs undergo only moderate global conformational changes upon multi-site phosphorylation. Thus, a more function-related question is how phosphorylation affects more locally the accessibility of the active site of an IDP. This is particularly important in light of the formation of "fuzzy complexes" [36]. Phosphorylation has in these cases been shown to inhibit or enhance transient interactions of the complex, for example by masking positively charged residues or by inducing cation-π interactions between aromatic and phosphorylated residues [37]. We here focus on the MD simulations for E-Cadherin and p130Cas with two phosphorylated residues (Table 1), which in both IDPs are located at interaction sites of binding partners and regulate their binding (p120 for E-Cadherin, and Src for p130Cas [38]). We analyzed the exposed solvent accessible surface area (SASA) of the binding site (comprising 10 residues including the two phosphorylation sites), i.e. the surface area of the residues involved in binding that is not shielded by other residues of the IDP. We observed that the binding sites of both E-Cadherin and p130Cas with the two phosphorylation sites at the binding sites present are significantly more exposed to the environment compared to the unphosphorylated states, with shielded areas approaching the values of the fully phosphorylated forms (Fig 7). These results imply a higher probability to interact with other binding partners. From this perspective, phosphorylation not only regulates binding by direct recognition of the phosphate-group(s) by the binding partner, but also by a reduction of the intramolecular shielding, in our specific cases by no less than 40% (Fig 7A and 7B). On the other hand, we speculate that stronger collapse by phosphorylation as observed for Ash1 or by dephosphorylation as observed for CTD2', p130Cas, and E-Cadherin, can protect the protein from undesired interactions, damage and degradation (Fig 7C) [39]. The V-shape changes in collapse propensity, according to our data for Ash1 and CTD2' (Fig 2), also prevail at higher salt concentrations so can indeed by relevant for protein-IDP interactions within the crowded and charged environment of the cell.

## Summary and conclusion

Here we have systematically studied the mechanism of IDPs phosphorylation by using extensive MD simulations of ~80-residue segments of four different IDPs, which feature up to 40% charged residues prior or after prior to phosphorylation. Our results suggest that care must be

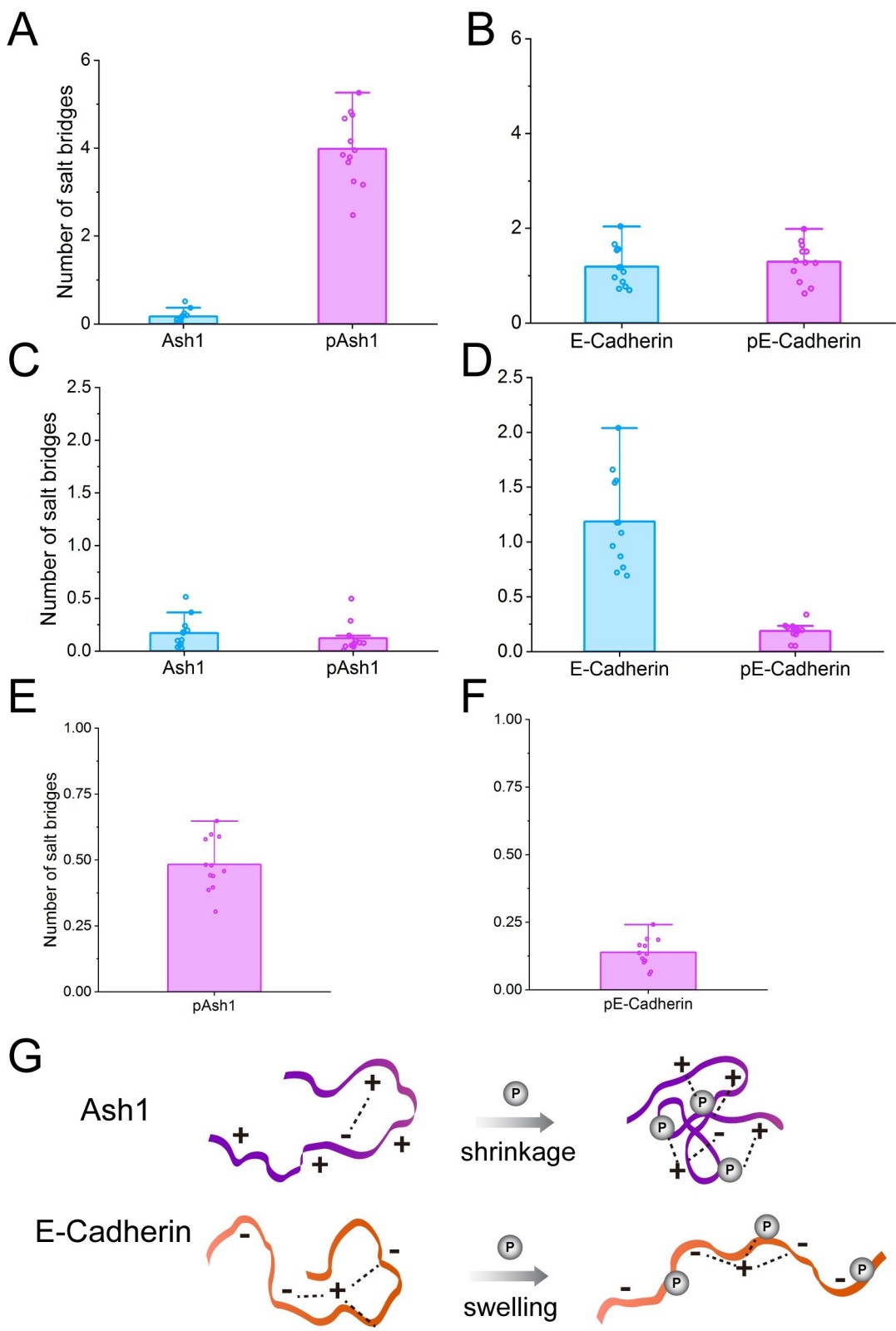

**Fig 6. Analysis of salt bridges.** The influence of phosphorylation on the average number of salt bridges in Ash1 and E-Cadherin (A and B) during MD simulations in 350 mM NaCl. The influence of phosphorylation on the only E/D–R/K salt bridges is

shown in C and D and the average salt bridges per phosphorylated residue are shown in E and F. Schematic view of how phosphorylation affects the salt bridges (dashed line) in Ash1 and E-Cadherin upon phosphorylation (G).

taken when studying IDPs with high abundance of charges in MD simulations even when using IDP-adequate force fields. A higher salt concentration to obtain reasonable conformational ensembles is not a perfect solution for the overstabilization of salt bridges, and future work is required to solve this problem by other parametrizations (e.g. polarizable IDP force-fields). This appears particularly critical when studying phase separation, where charged interactions play a key role [40]. We obtained moderate global and local conformational changes, which can be partly hidden in SAXS data by significant changes in the hydration shell of the IDPs upon phosphorylation. The global changes can be fully explained by the total net charge of the IDP, that is, by the overall electrostatic interactions. Based on our results, we predict more pronounced charge-induced changes in IDPs upon phosphorylation at lower screening, for example in experiments using minimal amounts of buffer and no additional salt. Our results have implications for the conformational control by phosphorylation (and analogously acetylation) of protein-protein recognition involving IDPs.

## Materials and methods

Molecular dynamics simulations were performed using GROMACS 2016.3 [41] to investigate the dynamics of different IDPs in wild type and phosphorylated states. The initial structures were modelled ab initio from sequence in an extended linear conformation by using Ambertools17 [42]. The N-terminus and C-terminus of the peptide were capped by an acetyl-capping group (ACE) and an amide-capping group (NME), respectively. The different phosphorylation states were modified by using PyTMs [43], a PyMoL [44] plugin for modeling common post-translational modifications based on wild type structures.

The system was described by using the AMBER99-sb*-ILDN force field [25] in combination with the TIP4PD water model [19] under particle mesh Ewald periodic boundary conditions. Parameters for phosphoserine, phosphothreonine and phosphotyrosine in different protonation states, in consistency with the AMBER force-field, are derived from Ref [45]. Each starting structure of different phosphorylation states of each of the investigated IDPs was placed in a dodecahedron box and solvated. System sizes varied between 0.5 million and 2 million atoms. The system was neutralized by adding ions, and extra NaCl was added to represent a specific ionic strength as indicated. First of all, the system was minimized using the steepest descent minimization approach. After the minimization, the system was equilibrated in the NVT ensemble with all-heavy atom restrained with a force constant of 1000 kJ/mol. The temperature was maintained at 300 K using a V-rescale thermostat with a coupling constant of 0.1 ps. Next stage of equilibration was carried on in the NPT ensemble also with all-heavy atom restrained with a force constant of 1000 kJ/mol, and where the pressure was maintained at 1 atmosphere using a Parrinello-Rahamn barostat with the coupling constant set to 2.0 ps. Both equilibrations were performed for 200 ps with a time step of 1 fs. For the 420 ns production run, the thermostat and barostat settings were the same as for the NPT run. To enable 2 fs time steps, bonds involving hydrogen atoms were constrained to equilibration length using the LINCS algorithm [46]. A real-space cutoff of 10 Å was used for the electrostatic and Lennard-Jones forces. Snapshots from each trajectory were stored every 50 ps. To further broadly sample the conformational space of IDPs, 7 new structures were extracted every 50 ns from 120 ns and served as initial structure of new simulations, aiming at convergent sampling. Each of these 7 structures were used as initial structure for new MD simulations using the same protocol and repeated for 2 replicas. In total, $14 \times 300$ ns = 4.2 μs production MD simulations were

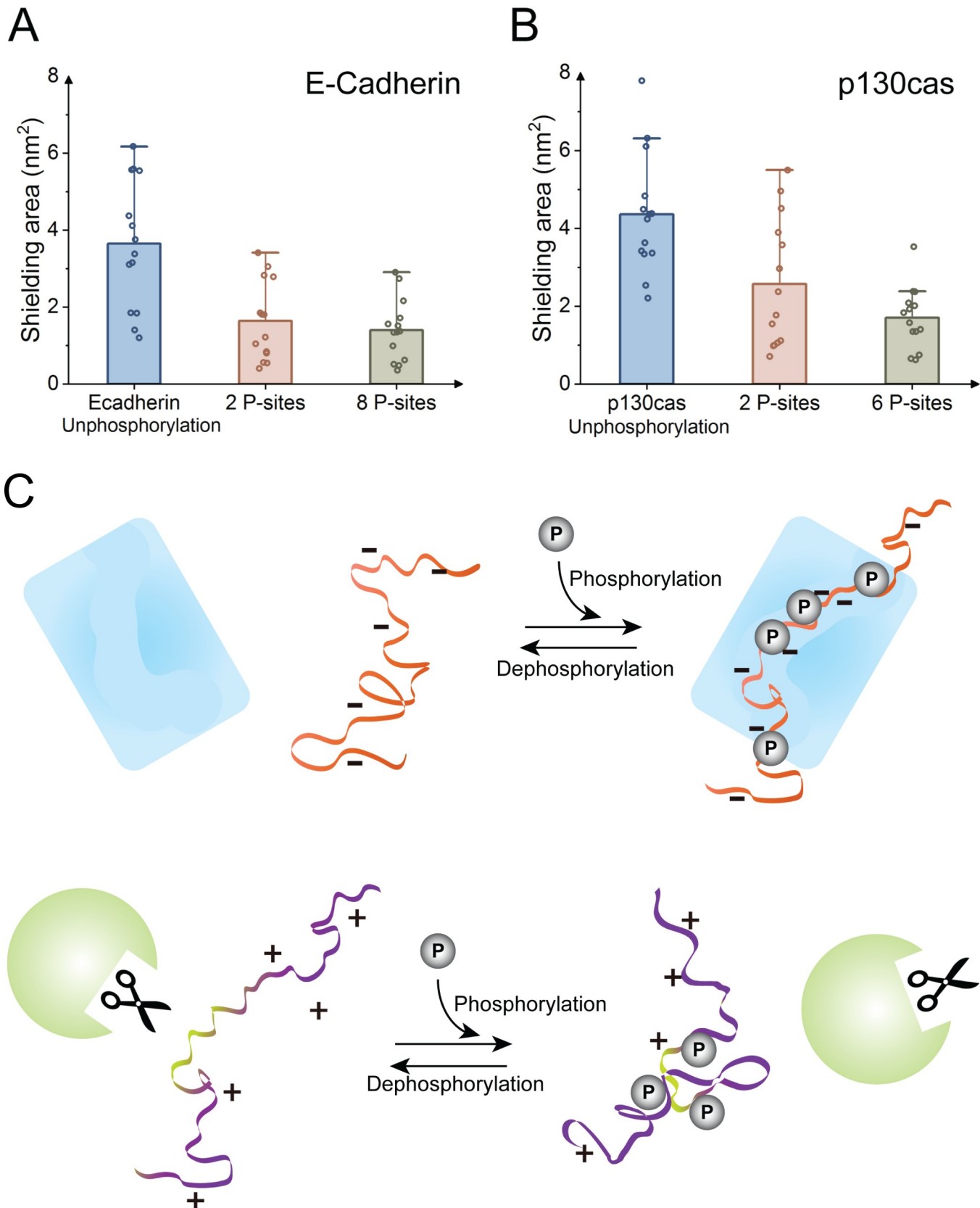

**Fig 7. Interaction site exposure of E-Cadherin and p130Cas changes upon phosphorylation and implications for biological functions of IDPs.** Shielded area of interaction protein residues by intramolecular interactions for E-Cadherin (A) and p130Cas (B) for unphosphorylated states, biologically relevant two phosphorylation sites (2 P-sites) and multiple phosphorylation sites (8 P-sites for E-Cadherin and 6 P-sites for p130Cas). Data points: average along single 400 ns trajectories; bars: average over 14 trajectories, errors bars: standard errors of the mean of the 14 averages. (C) Schematic view of a defense-attack perspective: phosphorylation can cause functional changes of IDPs beyond direct recognition. Phosphorylation around negatively charged active site can expose binding sites, resulting in an extended and favorable binding conformation that more easily interacts with binding partners, while phosphorylation of positively (or dephosphorylation of negatively) charged active sites can reduce exposure to the environment and protect IDPs e.g. from proteolysis.

performed for each phosphorylation state of different IDPs. The same settings were used for MD simulations with Charmm36m [22], a99SB-disp [23] and Amber99SB* ILDN-DERK [25,26] forcefields. In total, across all IDPs, phosphorylation states, and ionic strengths, 0.218 ms of aggregated MD simulations were performed. The simulations performed in this work are listed in S2 Table.

For sampling conformations of Ash1/pAsh1 and CTD'/pCTD' under different concentrations of NaCl, we initialized a 420 ns MD simulation from extended linear structures and randomly selected relatively extended conformations every 50 ns and they were used as new starting conformation for MD simulations, 3 x 400 ns in length for each NaCl concentration. Ten different concentration between 100 and 1500 mM were used.

Standard force fields are prone to over-stabilize salt bridges [24] and could not deal with polarization of atoms [47]. Here, by increasing the concentration of NaCl (see Results) and using Tip4p-D [19] solvent model, which strengthens the protein-solvent interactions, we partly overcame the limitations of standard protein force fields.

To get an estimate for the convergence of Rg, we analyze the Rg of Ash1 and CTD2' in their unphosphorylated and phosphorylated forms for different MD times, and observe reasonable convergence within 300 ns to 400 ns (S11 Fig).

To validate MD simulations and quantify differences between different SAXS analysis schemes, SAXS curves were reversely calculated using WAXSiS [34,18], which explicitly takes scattering from solvating water into account. For WAXSiS, additional MD simulations of only TIP4PD water 200 ns in length were performed as reference. SAXS curve fittings and respective $\chi^2$ values were computed using a popular chi-square statistic widely used in statistics and SAXS analyses (Equation (1) in Ref. [17]). Chemical shifts were calculated by using SHIFTX2 [48] every 200 ps and then averaged over the trajectories.

For production run, the first 100 ns simulations were regarded as equilibration and not used in analyses. All the other analyses were conducted by using GROMACS utilities, PyMol [44] or in-house scripts.

Numerical data for all figures are available in S1 Data.

## Supporting information

**S1 Fig. Global conformational changes in 100 mM NaCl of Ash1, measured by $R_G$, the mass-weighted radius of gyration, for the unphosphorylated (blue) form, the phosphomimetic version of Ash1 (all the phosphorylation sites replaced by Glu, cyan), and multi-site phosphorylated forms ($HPO_4^{-2}$ in red and $PO_4^{-2}$ in olive).**
(TIF)

**S2 Fig. Average number of salt bridges for Ash1/pAsh1 observed in MD simulations under different NaCl concentrations.**
(TIF)

**S3 Fig. Comparisons of back-calculated chemical shifts from MD simulations to NMR data under 150 mM and 400 mM NaCl as examples.** Comparisons of simulation-derived

(purple) and experimental chemical shifts (cyan) for unphosphorylated (left panel) and phosphorylated (right panel) forms of Ash1 show in-line agreement. Note that the experimental values for some residues were not available. Chemical shifts are for the atoms: A H, B Cα, C Cβ, D C, E N. Experimental data from REF [6].
(TIF)

**S4 Fig.** Global conformational changes in 100 mM NaCl, measured by $R_G$, the mass-weighted radius of gyration, of CTD2' for unphosphorylated (blue) and multi-site phosphorylated forms ($HPO_4^{-1}$ in red and $PO_4^{-2}$ in olive).
(TIF)

**S5 Fig.** Comparison of MD simulations to SAXS data. Residuals of fitting for Ash1 (A) and CTD2' (B) $\Delta = [I_{calc} (q) - I_{exp} (q)]$, q is the scattering vector. Guinier analyses were conducted for Ash1 (C and D) and CTD2' (E and F) for experimental and MD back-calculated SAXS curve.
(TIF)

**S6 Fig. Global conformational changes in 100 mM NaCl, measured by $R_G$, the mass-weighted radius of gyration, of Ash1 (A) and CTD2' (B) for unphosphorylated (blue) and multi-site phosphorylated forms ($PO_4^{-2}$ in olive) by using the Charmm36m forcefield.** The central structures of each state are shown in cartoon, with positively charged and negatively charged residues shown in blue and red, respectively. The phosphorylated residues are shown as orange sticks.
(TIF)

**S7 Fig.** Global conformational changes in 100 and 350 mM NaCl, measured by $R_G$, the mass-weighted radius of gyration, of Ash1 (A) and CTD2' (B) for unphosphorylated (blue) and multi-site phosphorylated forms ($PO_4^{-2}$ in olive) when using the a99SB-disp forcefield.
(TIF)

**S8 Fig.** Global conformational changes in 100 and 350 mM NaCl, measured by $R_G$, the mass-weighted radius of gyration, of Ash1 (A) and CTD2' (B) for unphosphorylated (blue) and multi-site phosphorylated forms ($PO_4^{-2}$ in olive) when using the Amber99SB* ILDN-DERK forcefield.
(TIF)

**S9 Fig.** Correlation of the absolute value of Net charge per residue (NCPR) with the radius of gyration, normalized to random coil dimension ($R_c = R_0 \times N^{0.588}$ [35], where $R_0$ is a constant and N is number of residues of IDP) for Ash1/pAsh1, CTD2'/pCTD2', E-Cadherin/pE-Cadherin and p130Cas/pp130Cas under 100 mM (left) and 350 mM (right) NaCl. Linear fits considering the error of Rg are shown as solid lines. Slopes and p-values for the linear correlations are shown as inset.
(TIF)

**S10 Fig. Correlation of parameters "Coulomb force" (A) and "Coulomb potential" (B) from sequence with radius of gyration, normalized to random coil dimension ($R_C = R_0 \times N^{0.588}$, where $R_0$ is a constant and N is number of residues of IDP) for Ash1/pAsh1 (light blue), CTD2'/pCTD2' (gray), E-cadherin/pE-cadherin (dark yellow) and P130cas/pP130cas (orange), respectively, under 350 mM (opaque) and 100 mM (partially transparent) NaCl.** Linear fits are shown in solid (350 mM NaCl) and dashed (100 mM NaCl) lines. $Q_i$ and $Q_i$ are charges at sequence positions i and j, |j-i| is their absolute residue distance along the

sequence.
(TIF)

**S11 Fig. Sampling factor and convergence of the MD simulations.** The radius of gyration
were compared for MD simulations of Ash1/pAsh1 and CTD2'/pCTD2' under 350 mM NaCl
of 50 ns, 100 ns, 200 ns and 300 ns simulation time. The first 100 ns simulations were regarded
as equilibration and not used in analyses.
(TIF)

**S1 Table. Root-mean-square deviation of simulation-derived (under different concentrations of NaCl) and experimental chemical shifts for unphosphorylated and phosphorylated
forms of Ash1.**
(PDF)

**S2 Table. Simulations performed in this work.**
(PDF)

**S1 Data. Numerical data for all figures.**
(ZIP)

## Acknowledgments

We thank Tanja Mittag for providing a new set of SAXS data on Ash1 and pAsh1 and Jannik
Buhr for help with statistical analyses.

## Author Contributions

**Conceptualization:** Fan Jin, Frauke Gräter.

**Data curation:** Fan Jin.

**Formal analysis:** Fan Jin, Frauke Gräter.

**Funding acquisition:** Fan Jin, Frauke Gräter.

**Investigation:** Fan Jin, Frauke Gräter.

**Methodology:** Fan Jin, Frauke Gräter.

**Project administration:** Fan Jin, Frauke Gräter.

**Supervision:** Fan Jin, Frauke Gräter.

**Validation:** Fan Jin, Frauke Gräter.

**Writing – original draft:** Fan Jin, Frauke Gräter.

**Writing – review & editing:** Fan Jin, Frauke Gräter.

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
