## [Decision Letter · Decision Letter 0]

14 Dec 2020

Dear Dr. Gräter,

Thank you very much for submitting your manuscript "How multisite phosphorylation impacts the conformations of intrinsically disordered proteins" for consideration at PLOS Computational Biology.

As with all papers reviewed by the journal, your manuscript was reviewed by members of the editorial board and by several independent reviewers. In light of the reviews (below this email), we would like to invite the resubmission of a significantly-revised version that takes into account the reviewers' comments.

We cannot make any decision about publication until we have seen the revised manuscript and your response to the reviewers' comments. Your revised manuscript is also likely to be sent to reviewers for further evaluation.

Sincerely,

Alexander MacKerell

Associate Editor

PLOS Computational Biology

Arne Elofsson

Deputy Editor

PLOS Computational Biology

Reviewer's Responses to Questions

**Comments to the Authors:**

Reviewer #1: Very timely problem, addressed in a very rigorous manner. Owing to the relevance of the phenomenon studied and the number of not sufficiently careful computational studies on the field, the reviewer is grateful to the authors that they 'pour clean water into the glass'.

Minor comments:

- For general audience, it would be useful to prepare a scheme summarising the effect of phosphorylation in disordered or partly structured domains based on your findings and previous results of others. Can you elaborate the potential mechanisms in the intro, as well?

- The impact of phosphorylation also depends on to what extent the ID chains get ordered upon binding. Can you elaborate this aspect, in particular in the lights of how phosphorylation can screen protein-protein interactions (autoinhibitory effects)?

I can give a long list of citations of experimental studies eg. on Max, Ets-1, Ire-1, FACT etc., summarised in reviews on fuzzy complexes.

- I appreciate the large-scale study on different FFs. Can you detail the polarisation effect, solvent models and maybe sampling factors?

- Application to biomolecular condensates. It is good that you mention this, but would be worth to mentioning in intro. Not only dynamic condensates but formation of many different kinds of higher-order assemblies are regulated by phosphorylation (see Wu& Fuxreiter Cell 2016). The question is then, how the screening effect of phosphorylation which you describe relates to the interactions in 'traditional' assemblies and protein droplets. This is a crucial point to allocate the sites of regulation for biomolecular condensates.

Reviewer #2: In their manuscript “How multisite phosphorylation impacts the conformations of

intrinsically disordered proteins”, Jin and Gräter describe a systematic study in which they use extensive molecular dynamics simulations to characterize the effects of phosphorylation on the structure of IDPs of different amino acid composition. From Table 2 of the SI (list of simulations), it is clear that their study has covered a variety of sequences (Ash1, E-Cadherin, CTD2’ and p130Cas), phosphorylation states, salt concentrations and force fields, for a total sampling time of 0.272 ms. The manuscript is clear and understandable, and the authors have carried out a careful study on an important and current problem in the field of IDPs.

I have some suggestions for revision that could strengthen the manuscript and some questions about details in the manuscript:

- In the introduction, it would be relevant to mention recent work from the Skepö lab (https://pubs.acs.org/doi/abs/10.1021/acs.jctc.9b01190) on the effect of phosphorylation on the conformation of another IDP - statherin. This work looks at a similar problem as this manuscript.

- Table 1 — it would be helpful to provide the Genbank accession numbers for the sequences because it is unclear what species they are from.

- On p. 8 line 145, the authors cite ref. 25 for the statement “Given that the current protein force fields have been shown to overstabilize electrostatic interactions“, but as far as I can tell, reference 25 doesn’t support that statement. Similarly in lines 150-152.

- The authors suggest that salt bridges are overstabilized based on differences in Rg (p. 8, line 150-152). Rg is an global feature. This statement could be supported quantitatively by providing populations of salt bridges in the simulations with differing salt concentrations.

- When comparing to SAXS/NMR measurements, it would be useful to provide the salt concentrations in these measurements.

- On p. 9, the authors state that “We conclude that using 350mM NaCl yields conformational ensembles of the two IDPs and their phosphorylated forms which agree with SAXS and NMR data and thus resemble the experimental ensemble.” Is 350mM the concentration for which the agreement is best? Do the SAXS and NMR data both show the best agreement at the same concentration?

- What method was used to compute the chemical shifts? Can the authors provide error estimates for these shifts in Fig. 2? They state that the agreement is “very good” but without reporting error or the method used to compute the shifts, it is hard to assess this. Which salt concentration was used in the simulations reported in Fig. 2?

- p. 11, lines 191-193: The authors’ assessment of the accuracy of the C36m force field is problematic. They have carried out only a single simulation of 180 ns for 4 systems (~30 times less simulation than other systems). Despite the short simulation, they claim “In the case of Charmm36m, irrespective of the IDP and the phosphorylation, we observe a strong over collapse of the IDP which is not in line with the SAXS data and dominate over the charge effect.” This statement isn’t supported by the data.

- The paper would benefit from a systematic analysis of salt bridges, rather than relying on Rg (a global structural property) to infer effects on local structure, like salt bridges.

- p. 14, line 234: The authors refer to “the hydrodynamic radius”, but no RH is reported, so I think this must be a typo.

- p. 15, line 259: The authors suggest that the plot shown in Fig. 4 has a V-shape, but given the large error, this isn’t well-supported by the results.

- Fig. 5 is unclear and difficult to interpret.

- It is unclear how the equilibration of the simulations was treated. E.g. in Fig. S4, the authors exclude 40 ns for equilibration, but the reason for this decision wasn’t clear. Was 40 ns used for all systems? If so, why?

- Have the authors compared to all available experimental data for these systems?

**Have all data underlying the figures and results presented in the manuscript been provided?**

Reviewer #1: Yes

Reviewer #2: **No: **Numerical data for the figures has not been provided.

PLOS authors have the option to publish the peer review history of their article (what does this mean?). If published, this will include your full peer review and any attached files.

Reviewer #1: No

Reviewer #2: No
---

## [Decision Letter · Decision Letter 1]

6 Apr 2021

Dear Dr. Gräter,

We are pleased to inform you that your manuscript 'How multisite phosphorylation impacts the conformations of intrinsically disordered proteins' has been provisionally accepted for publication in PLOS Computational Biology.

Best regards,

Alexander MacKerell

Associate Editor

PLOS Computational Biology

Arne Elofsson

Deputy Editor

PLOS Computational Biology

Reviewer's Responses to Questions

**Comments to the Authors:**

Reviewer #1: The authors have addressed all the comments and the paper was very good and solid anyway.

Reviewer #2: The authors have addressed the comments of both reviewers. To reiterate my first review, the authors' study is carefully done and represents an important contribution to the field of disordered proteins. I recommend that it be published.

**Have the authors made all data and (if applicable) computational code underlying the findings in their manuscript fully available?**

Reviewer #2: None

PLOS authors have the option to publish the peer review history of their article (what does this mean?). If published, this will include your full peer review and any attached files.

Reviewer #1: No

Reviewer #2: No

**Have all data underlying the figures and results presented in the manuscript been provided?**

Reviewer #1: Yes

---

## [Editor Report · Acceptance letter]

26 Apr 2021

PCOMPBIOL-D-20-02013R1 

How multisite phosphorylation impacts the conformations of intrinsically disordered proteins

Dear Dr Gräter,

I am pleased to inform you that your manuscript has been formally accepted for publication in PLOS Computational Biology. Your manuscript is now with our production department and you will be notified of the publication date in due course.

With kind regards,

Katalin Szabo
